# Biological Safety Evaluation and Surface Modification of Biocompatible Ti–15Zr–4Nb Alloy

**DOI:** 10.3390/ma14040731

**Published:** 2021-02-04

**Authors:** Yoshimitsu Okazaki, Shin-ichi Katsuda

**Affiliations:** 1Department of Life Science and Biotechnology, National Institute of Advanced Industrial Science and Technology, 1-1 Higashi 1-Chome, Tsukuba 305-8566, Ibaraki, Japan; 2Japan Food Research Laboratory, 2-3 Bunkyo, Chitose 206-0025, Hokkaido, Japan; katudas@jfrl.or.jp

**Keywords:** Ti–15Zr–4Nb alloy, biological safety evaluation, ISO 10993 series, accelerated extraction, grit blasting, osteocompatibility, morphometrical parameters, maximum pullout load

## Abstract

We performed biological safety evaluation tests of three Ti–Zr alloys under accelerated extraction condition. We also conducted histopathological analysis of long-term implantation of pure V, Al, Ni, Zr, Nb, and Ta metals as well as Ni–Ti and high-V-containing Ti–15V–3Al–3Sn alloys in rats. The effect of the dental implant (screw) shape on morphometrical parameters was investigated using rabbits. Moreover, we examined the maximum pullout properties of grit-blasted Ti–Zr alloys after their implantation in rabbits. The biological safety evaluation tests of three Ti–Zr alloys (Ti–15Zr–4Nb, Ti–15Zr–4Nb–1Ta, and Ti–15Zr–4Nb–4Ta) showed no adverse (negative) effects of either normal or accelerated extraction. No bone was formed around the pure V and Ni implants. The Al, Zr, Nb, and Ni–Ti implants were surrounded by new bone. The new bone formed around Ti–Ni and high-V-containing Ti alloys tended to be thinner than that formed around Ti–Zr and Ti–6Al–4V alloys. The rate of bone formation on the threaded portion in the Ti–15Zr–4Nb–4Ta dental implant was the same as that on a smooth surface. The maximum pullout loads of the grit- and shot-blasted Ti–Zr alloys increased linearly with implantation period in rabbits. The pullout load of grit-blasted Ti–Zr alloy rods was higher than that of shot-blasted ones. The surface roughness (*Ra*) and area ratio of residual Al_2_O_3_ particles of the Ti–15Zr–4Nb alloy surface grit-blasted with Al_2_O_3_ particles were the same as those of the grit-blasted Alloclassic stem surface. It was clarified that the grit-blasted Ti–15Zr–4Nb alloy could be used for artificial hip joint stems.

## 1. Introduction

The biocompatibility of various metals and alloys has been specifically examined by Steinemann [1]. The adverse effects of metal ions include the cytotoxicity of trace amounts of vanadium (V) and aluminum (Al) ions in Ti–6Al–4V alloy and the allergy-related characteristics of nickel (Ni) ions [2,3,4]. Recently, many types of Ti alloy containing noncytotoxic elements such as zirconium (Zr), niobium (Nb), tantalum (Ta), and molybdenum (Mo) have been developed for various medical applications [5,6,7,8,9,10,11,12,13,14,15,16]. Ti–15Zr–4Nb–(0 to 4) Ta alloy, which is an alpha (α)–beta (β)-type alloy, has been developed in Japan as a highly biocompatible alloy [17], and it is standardized in JIS T 7401-4 [18]. As the addition of Ta to Ti alloys increases the manufacturing cost, the microstructure and mechanical properties of Ta-free Ti–15Zr–4Nb alloy have been investigated [17].

The International Organization for Standardization (ISO) 10993 series standardizes the biological safety evaluation required for application for approval to market medical devices. In this series, various test methods for metallic medical devices implanted into the human bone, such as metallic orthopedic implants (permanent contact), are necessary for the following evaluation items: cytotoxicity [19], sensitization [20], irritation/intracutaneous reactivity [19], material-mediated pyrogenicity [20], acute systemic toxicity [21], subacute toxicity [21], genotoxicity [22,23], carcinogenicity [22], and implantation [24]. In particular, evaluation of the biological safety of metal ions is important for metallic implants. 

Physiological saline (0.9 mass% NaCl) solution is commonly used to extract metal ions in the ISO 10993 series. On the other hand, dilute hydrochloric acid physiological saline (0.9%NaCl + HCl) solution (0.9%NaCl solution adjusted to pH 2 with hydrochloric acid (HCl)) is specified in ISO 16428 [25] as an accelerated (exaggerated) extraction solution for evaluation of corrosion resistance. It is expected that this accelerated extraction solution will be applied to biological safety evaluation. We focused on biological safety evaluation by the accelerated extraction of Ti–Zr alloys in accordance with the ISO 10993 series. We carried out histopathological analysis of the long-term implantation of pure metals (e.g., V, Al, Ni, Zr, Nb, and Ta) as well as Ni–Ti and high-V-containing Ti–15V–3Al–3Sn alloys.

Shot peening and grit blasting techniques are used in several surface modifications in artificial hip arthroplasic devices, such as the cementless Alloclassic Zweymüller Stepless (SL) stem (Zimmer Biomet, Tokyo, Japan). This stem is made of a V-free Ti–6Al–7Nb alloy [26]. A surface roughness (*Ra*) of 3 to 5 μm on average is obtained by grit blasting the entire prosthetic surface with 24-grit (blast particle size: 750 μm) highly pure Al_2_O_3_ corundum particles [27]. The national joint replacement registries of blasted Ti femoral stems in cementless primary total hip arthroplasty show satisfactory clinical results of these stems [28,29,30]. Focusing on grit blasting, we developed a grit blasting technique using Ti–Zr alloys.

In this study, to develop orthopedic Ti–Zr alloy implants with excellent biocompatibility and osseointegration, biological safety evaluation tests of Ti–Zr alloys in accordance with the ISO 10993 series were performed under both normal and accelerated extraction conditions. Furthermore, to compare the osteocompatibility of implantable alloys and investigate the effect of metal ions, we conducted histopathological analysis using morphometrical parameters for long-term rat implantation tests of pure V, Al, Ni, Zr, Nb, and Ta metals as well as Ni–Ti and high-V-containing Ti–15V–3Al–3Sn alloys. The effect of the dental implant shape on morphometrical parameters was investigated using rabbits. Moreover, we examined the maximum pullout properties of grit-blasted Ti–Zr alloys after their implantation in rabbits. We believe that the results obtained in this study will be useful for the development of various orthopedic implants and regulatory approval applications for new orthopedic implant devices. In addition, a method of evaluating biological safety under accelerated extraction condition is useful for evaluating the biological safety of highly biocompatible, biodegradable and other materials.

## 2. Experimental Method 

### 2.1. Test Specimens

Table 1 shows the chemical compositions of the three Ti–Zr alloys used for biological safety evaluation tests in accordance with the ISO 10993 series. To examine the effect of the added Ta on the biological safety of Ti–Zr alloys, Ta-free Ti–15%Zr–4%Nb (Ti–15–4), Ti–15%Zr–4%Nb–1%Ta (Ti–15–4–1), and Ti–15%Zr–4%Nb–4%Ta (Ti–15–4–4) alloys were used for various biological safety evaluation tests. The chemical composition of the Ti–6%Al–4%V (Ti–6–4) alloy used in the implantation test for comparison is shown in Table 1 [17] (here and hereafter, values in alloy compositions indicate mass%). Plate specimens with dimensions of 20 mm × 20 mm × 1 mm (thickness) each were cut from these three Ti–Zr alloys for extraction in the biological safety test. In addition, in the direct contact test for cytotoxicity, disk specimens (diameter: 3.5 mm; thickness: 1 mm) were cut from the Ti–15–4–4 alloy. The surface of each plate was polished with 1000-grit waterproof emery paper. For rat implantation, specimens with a diameter of 1.2 mm and a length of 2.5 mm were used. Each specimen was ultrasonically cleaned in ethanol.

The surface roughness (*Ra*)of the specimen surface was measured with Surfcom 130A equipment (ACCRETECH, Tokyo, Japan) after machining the test specimens. To compare with the osteocompatibility of implantable alloys for long-term implantation in rats, specimens with a diameter of 1.2 mm and a length of 2.5 mm were machined with high-purity metals (Nilaco Co., Tokyo, Japan: V (99.8% purity; *Ra*, 0.46 μm), Al (99.99% purity; *Ra*, 0.22 μm), Ni (99.7% purity; *Ra*, 0.19 μm), Zr (99.7% purity; *Ra*, 0.21 μm), Nb (99.9% purity; *Ra*, 0.30 μm), and Ta (99.95% purity; *Ra*, 0.31 μm)), Ni–Ti alloy (Kobe Steel, Ltd., Hyogo, Japan; Ni, 55.5 mass%; *Ra*, 0.24 μm), and high-V-containing Ti–15%V–3%Al–3%Sn (Ti–15–3–3) alloy (Kobe Steel, Ltd., Hyogo, Japan; *Ra*, 0.25 μm). Dental implants (Degusse AG Dental, Hanau, Germany; diameter, 5 mm; length, 14 mm) and smooth rod specimens (diameter, 5 mm; length, 14 mm; *Ra*, 0.07 μm) made of Ti–15–4–4 alloy were manufactured for the rabbit implantation test. The surface of the threaded portion of the dental implant was shot-blasted.

To investigate the effect of blasting on maximum pullout properties after rabbit implantation, Ti–15–4 alloy plates (width, 3 mm; length, 15 mm; thickness, 1 mm) were cut from surfaces of Ti–Zr alloy plates grit-blasted with 24-grit (central particle size, 600–710 μm) highly pure (≥99%) Fuji Random WA Al_2_O_3_ particles (Fuji Seisakusho Co., Ltd., Tokyo, Japan). Plates of the same size were cut from the Alloclassic-SL stem surface as flat as possible to compare the pullout load of grit-blasted Ti–Zr alloy plates. A hole of 1 mm diameter was drilled at one end of the plate. The specimen surface after grit blasting was ultrasonically cleaned in ethanol. The two grit-blasted plates were bonded together with medical grade adhesive (Aron Alpha A “Sankyo”, Daiichi Sankyo, Tokyo, Japan) so that both the front and back surfaces were grit-blasted. Ti–15–4–4 alloy rods (diameter, 3 mm; length, 20 mm) were machined (*Ra*, 0.39 ± 0.06 μm) and shot-blasted (*Ra*, 0.51 ± 0.01 μm) with high-purity Al_2_O_3_ particles after machining for comparison. To investigate the effect of removing residual Al_2_O_3_ particles, the grit-blasted surface was chemically etched in 2 vol% nitric acid aqueous solution containing 1 vol% hydrogen fluoride for up to a maximum of 4 min. 

### 2.2. Animals, Cells, and Bacteria

Mice, rats, guinea pigs, and rabbits were used for animal studies. Mice of the Slc:ICR strain, rats of the BrlHan:WIST@Jcl (GALAS) strain, guinea pigs of the Slc:Hartly strain, and Japanese white rabbits (Kbs:JW) were obtained from Japan SLC Inc. (Shizuoka, Japan), CLEA-Japan Co. (Tokyo, Japan), Japan SLC Inc. (Shizuoka, Japan), and Kitayamarabesu Co., Ltd. (Nagano, Japan), respectively. They were acclimated for four to five days and maintained at 20–26 °C and 30–80% relative humidity at a 12 h light–dark cycle. During the experimental periods, five mice were housed in a plastic cage (225 W × 338 D × 140 H mm), three rats in a plastic cage (225 W × 338 D × 140 H mm), one to three guinea pigs in a fiber reinforced plastics (FRP) cage (260 W × 382 D × 200 H mm), and one rabbit in an FRP cage (350 W × 527 D × 350 H mm). The CFR-1 diet (gamma-ray-irradiated, Oriental Yeast Co., Ltd., Kanagawa, Japan) was fed to mice and rats, and the LRC4 diet (gamma-ray-irradiated, Oriental Yeast Co., Ltd., Kanagawa, Japan) was fed to guinea pigs and rabbits. Tap water that met the requirements of “potable water ” in Specifications and Standards for Foods, Food Additives, etc. (Ministry of Health and Welfare Notification, No. 370, 1959), as determined by periodic inspection, was given *ad libitum*. Animal species, number, sex, and body weight were in accordance with the designation of guidelines. During implantation, animals were under general anesthesia. When the observation period was finished, animals were humanely killed under general anesthesia. Regarding the use of animals, we followed the “Chitose Institute Animal Experiment Ethics Code” in Japan Food Research Laboratory (JFRL) on the basis of the Act on Welfare and Management of Animals (Act No. 105), Standards Relating to the Care and Keeping and Reducing Pain of Laboratory Animals (Notice of the Ministry of the Environment No. 88), and related guidelines.

A V79 cell line obtained from the Cell Bank (National Institutes of Biomedical Innovation, Health and Nutrition, Osaka, Japan) and a Chinese hamster lung (CHL/IU) cell line obtained from Pharma Biomedical Co., Ltd. (Osaka, Japan) were used for cytotoxicity and chromosomal aberration tests. For the bacterial reverse mutation test, four strains of *Salmonella typhimurium* (TA98, TA100, TA1535, and TA1537) and *Escherichia coli* (WP2uvrA) were obtained from Japan Bioassay Research Center, Kanagawa, Japan.

### 2.3. Biological Safety Evaluation Tests

Biological safety evaluation tests were performed at Chitose Laboratories, JFRL (Hokkaido, Japan). All biological safety tests were conducted in accordance with the principles of good laboratory practice (GLP), and we followed the Ordinance of the Japanese Ministry of Health, Labor, and Welfare (MHLW) No. 37 (23 March 2005) concerning the standards for conducting nonclinical tests on the safety of medical devices. Regarding the basic principles of biological safety evaluation required for application for approval to market medical devices, we conformed to the annex “Basic Principle for Biological Safety Evaluation of Medical Devices” and the attachment “Guidance on Test Methods for Biological Safety Evaluation of Medical Devices” of Notification 0301 No. 20 of the Pharmaceutical and Food Safety Bureau (1 March 2012). 

Biological safety evaluation tests of three Ti–Zr alloys were performed under normal extraction condition in accordance with the ISO 10993 series and under accelerated extraction condition developed in this study. The conditions of biological safety evaluation tests are shown in Table 2. In a cytotoxity test, the use of a culture medium with serum is preferred for extraction because it supports cellular growth as well as enables the extraction of both polar and nonpolar substances. For normal extraction, plate specimens were immersed in the 0.9%NaCl solution at 121 ± 2 °C for 1 ± 0.1 h. The accelerated extraction solution (0.9%NaCl + HCl, pH 2) was prepared as follows. Hydrochloric acid (1 mol/L) was added to physiological saline (0.9%NaCl) solution, and the mixture was adjusted to pH 2 (0.9%NaCl + HCl solution). All test specimens were ultrasonically cleaned in ethanol and sterilized in an autoclave at 121 °C for 15 min. After drying, the test specimens were immersed for 7 d (168 ± 2 h) in the 0.9%NaCl + HCl solution at 37 ± 1 °C for extraction. After sufficient stirring, the pH of the 0.9%NaCl + HCl solution after extraction was neutralized to 6 ± 1 using 1 and 0.1 mol/L NaOH solutions to obtain test extracts for various biological evaluation tests. Blank extracts (blank control) were similarly prepared but without the Ti–Zr alloy. A test-specimen-treated group and negative (blank control and so on) and positive control groups were established under different conditions necessary for each biological safety evaluation test to confirm the sensitivity of the test systems.

Morphometrical parameters for estimating the formation rate of new bone were examined by the implantation of stainless steel, Co–Cr–Mo, Ti–6Al–4V, and Ti–15Zr–4Nb–4Ta alloys into the rat femur and tibia for up to 48 weeks. Four morphometrical parameters are suggested to be useful for estimating osteocompatibility: (1) new bone formation rate (%) = (total length of new bone formed around implant)/(length of surrounding implant existing in bone marrow) × 100, (2) bone contact rate (%) = (total length of new bone in direct contact with implant)/(length of surrounding implant in bone marrow) × 100, (3) new bone thickness = (total area of new bone)/(total length of new bone formed around metal implant), and (4) osteoid formation rate (%) = ((total area of osteoid)/total area of new bone (osteoid plus calcified bone)) × 100 [31,32]. In this study, bone formation, bone contact, and osteoid formation rates were compared between test and control specimens.

### 2.4. Implantation of Pure Metals and Ni–Ti and High-V-Containing Ti Alloys in Rats

The test specimens (diameter, 1.2 mm; length, 2.5 mm) of pure V, Ni, Al, Zr, Nb, and Ta metals as well as Ni–Ti and Ti 15–3–3 alloys were implanted into 68 rat femurs for 4, 12, 26, and 52 weeks. Undecalcified thin vertical sections were stained with the Villanueva–Goldner stain. Bone formation rate, bone contact rate, and new bone thickness (μm) were measured for each specimen. 

### 2.5. Implantation of Dental Implant and Grit-Blasted Ti–Zr Alloy Rods in Rabbits 

The effects of the dental implant (screw) shape on morphometrical parameters and the maximum pullout properties of grit-blasted Ti–Zr alloys were investigated by implantation in rabbits. Figure 1 shows a schematic illustration of the method of implantation into the femur of rabbits. A dental implant (Ti–15–4–4), grit-blasted Ti alloys (Ti–15–4 and Ti–6–7), shot-blasted Ti–15–4–4, and smooth-surface and machined Ti–15–4 alloy rods for comparison were implanted into the distal epiphysis of the femur. For histological analysis of the dental implant and smooth-surface Ti–15–4 rods, undecalcified thin sections were subjected to Villanueva–Goldner staining. The rate of bone formation (%) around the screws was compared with that around the smooth rods. After the grit-blasted, shot-blasted, and machined Ti–15–4 rods were implanted for 4, 8, 12, 16, and 24 weeks, the pullout test was carried out at a crosshead speed of 0.5 mm/min. Maximum pullout loads were determined from load–displacement curves.

### 2.6. Statistical Analyses

Mean values and standard deviations were calculated for at least eight specimens in each pullout test. Statistical analyses for specimens and the blank test group were conducted using the data analysis software SPSS for Windows (SPSS Japan Inc., Tokyo, Japan). The level of significance was set at 5%. The data analyzed were body weight, the amount of food intake, urinalysis results, blood test results, and weights of organs. The urinalysis results obtained were compared among the groups by the Mann–Whitney tests. For the measurement of body weight, a test of homoscedasticity was performed for each extraction medium; the groups were compared at the level of significance of 5% by Student’s *t*-test for homoscedastic variance and by Welch’s test for heteroscedastic variance. The van der Waerden method was used to calculate the half-maximal inhibitory concentration (IC_50_) in the cytotoxicity test. 

## 3. Results and Discussion

### 3.1. Biological Safety Evaluation Test

Table 3 shows the biological safety evaluation test results obtained with Ti–15–4, Ti–15–4–1, and Ti–15–4–4 alloys. The effects of normal (in 0.9%NaCl and medium)- and accelerated (in 0.9%NaCl+HCl; pH 2)-condition extracts were compared using Ti–15–4 alloys. All of the tests performed in accordance with the ISO 10993 series showed no effect (negative) of either extract. No significant differences were found between the groups in the implantation tests.
(1)In the cytotoxicity (colony formation) tests at six concentrations (3.13, 6.25, 12.5, 25, 50, and 100%) of specimen extracts in the medium, no decrease in the rate of colony formation (93–101%) was induced by the test Ti–15–4 alloy. This result indicates that the test specimens were noncytotoxic.(2)In the maximization tests for sensitization evaluation, an average score of 1 or higher was considered positive for skin reactions in accordance with the ISO 10993-10 criteria. The average scores for the accelerated-condition (in 0.9%NaCl + HCl) and normal-condition (in 0.9%NaCl ) extracts were 0 for the test specimens subjected to 24 and 48 h treatments, indicating that the test specimens caused no sensitization response (no erythema) to the skin.(3)In the irritation tests of rabbits, the accelerated- and normal-condition (in 0.9%NaCl) extracts of 0.2 mL each were intradermally injected. No erythema or edema was observed at any of the injection sites at any of the observation times in all groups, resulting in a score of 0 for intradermal reactions. No significant differences in scores were observed among the accelerated condition extract injected, normal condition extract injected, and blank extract injected groups, indicating that the test specimens caused no intradermal reaction in the rabbits.(4)In the systemic toxicity tests, for the acute systemic toxicity of accelerated- and normal-condition extracts, no effects of the extracts on the general conditions and the weight of any of the mice and no abnormalities in either intraperitoneal or intrapleural organs were observed. This result indicates that the accelerated- and normal-condition extracts from the test specimens had no acute systemic toxicity. As for the normal-condition extract (121 °C for 1 h) intravenously injected to rats for 21 days, there were no significant differences between the test and control groups for both male and female rats in the weight and the amount of food intake or in the results of the urine test. In the weights of organs and the results of the hematologic and blood biochemical tests, negligible differences that reached statistical significance were observed for some items, but they were considered to be of no toxicological significance. In the macroscopic examination of the systemic organs, no abnormalities were observed for both male and female rats. In the histopathological examination, changes were occasionally found, but no noteworthy abnormalities were observed for both male and female rats. The above results indicated that no clear systemic toxicity was expressed when the normal-condition (in 0.9%NaCl) extracts from the test specimens were intravenously injected to male and female rats once a day for 21 days.(5)In the genotoxicity tests, for gene mutation inducibility, no increase in the number of revertant colonies was found. The Ti–15–4 alloy immersed in the accelerated extraction 0.9%NaCl + HCl solution showed no gene mutation inducibility. As for chromosomal aberration inducibility, no increase in the frequency of appearance of cells with chromosomal aberrations was found. This indicated that the Ti–15–4 alloy did not induce chromosomal aberrations.(6)In the implantation tests, no cellular infiltration was observed around the test specimen, and no degeneration, necrosis, bleeding, or other tissue reactions were found. In the histopathological examination, the formation of new bone was observed around the test specimen; the new bone was in direct contact with the test specimen and was calcified for all rats. Similar reactions were observed for the sites where the control Ti–6–4 was implanted. The Ti–15–4 alloy was not inflammatory but osteoconductive, similar to the control Ti–6–4.

Similar negative findings were obtained in many studies on the biological evaluation of various Ti alloys [33,34,35,36,37]. The concentration of Ti in extracts was determined (ng/mL) by inductively coupled plasma mass spectrometry (ICP-MS, NexION 300D, PerkinElmer, Kanagawa, Japan). No Ti or alloying elements were detected in the extracts of Ti–15–4, Ti–15–4–1, and Ti–15–4–4 alloys immersed in the 0.9%NaCl solution. The Ti concentrations in extracts of Ti–15–4, Ti–15–4–1, and Ti–15–4–4 alloys immersed in Eagle’s minimal essential medium (EMEM) were 0.02, 0.01, and 0.02 μg/mL, respectively. On the other hand, the Ti concentrations of Ti–15–4, Ti–15–4–1, and Ti–15–4–4 alloys immersed in 0.9%NaCl + HCl were 0.4, 0.27, and 0.32 μg/mL, respectively. It is considered that this absence of effect found in biological safety tests was due to the small amount of Ti ions released. This is because the Nb_2_O_5_-, ZrO_2_-, and Ta_2_O_5_-containing TiO_2_ oxide film that formed on the Ti alloy surface inhibited the release of metal ions [38,39]. 

From these results, it was clarified that the biological safety can be evaluated in accordance with the ISO 10993 series using extracts obtained with the accelerated extraction 0.9%NaCl + HCl solution. In a previous study, skin sensitization to Ni, chromium (Cr), Ti, and Zr has been evaluated by administering a high-concentration metal salt solution to animals [40]. Metal salts may contain impurities different from those in alloys, and exposure at high concentrations in biological safety tests does not reflect the actual situation of exposure to alloys used clinically. Furthermore, our previous study has shown that extraction with 0.9%NaCl is a mild extraction condition for simulating exposure to various alloys, including Ti–15–4–4 exposed to various body fluid components [41]. We used the accelerated extraction condition in this study and found that the Ti concentration of the Ti–15–4–4 alloy under this condition was the same as that of the alloy implanted in the rat tibia for 48 weeks (approximately 0.5 μg/g rat tibia) in our previous study [42]. Thus, the accelerated extraction condition mimics the in vivo condition well and is useful for evaluating the biological safety of highly biocompatible materials, biodegradable materials, and so on [43,44,45].

### 3.2. Osteocompatibility of Pure Metals and Ni–Ti and High-V-Containing Ti Alloys

Figure 2 shows the optical micrographs of the undecalcified bone sections subjected to Villanueva–Goldner staining at 52 weeks after implantation with pure V, Ni, Al, Zr, and Nb metals and Ni–Ti alloy. No bone was formed around the pure V and Ni implants. The Al, Zr, Nb, and Ni–Ti implants were surrounded by new bone. The new bone formed around the Al implant was a thin layer. 

Osteocompatibilities obtained using the morphometrical parameters for pure V, Ni, Al, Zr, Nb, and Ta metal as well as Ti alloy (Ni–Ti, Ti–15–3–3, Ti–6–4, and Ti–15–4–4) implants are shown in Figure 3 and Figure 4. The results with the same marks shown in Figure 3a–c and Figure 4a–c were obtained under the same test conditions using the same test specimens. As shown in Figure 3, no new bone was formed on pure Ni during the entire implantation period. For pure V, three bone parameters decreased in the early stage of implantation, whereas for pure Al, they tended to increase with the implantation period because an oxide film was formed on the pure Al plate surface, which inhibited the release of Al ions. 

The new bone formation rate was approximately 100% in the early implantation period (4 weeks) for pure Zr, Nb, and Ta implants (Figure 3a). The rate of bone contact was higher on pure Zr than on pure Nb and Ta. The new bone thicknesses for pure Zr, Nb, and Ta implants were approximately 40 μm 12 weeks after implantation. The difference in new bone thickness among the Zr, Nb, and Ta implants was small. The results of Ti–15–4–4 and Ti–6–4 alloys, shown in Figure 4, were taken from the literature [32]. The new bone surrounding Ti–Ni and high-V-containing Ti–15–3–3 alloys tended to be slightly thinner than that surrounding Ti–15–4–4 and Ti–6–4 alloys. The bone contact rates of these alloys were lower in the early and long implantation periods. The bone formation rate of the Ti–15–3–3 alloy was low 4 weeks after implantation. The bone forming capability of pure Zr, Nb, and Ta metals was similar to that reported in [46,47,48]. An evaluation of osseointegration using similar bone morphometrical parameters as in this study has been previously performed [49]. These results support those of material development of Ti alloys to which Zr, Nb, and Ta have been added [5,6,7,8,9,10,11,12,13,14,15,16].

### 3.3. Rate of Bone Formation on Dental Implant in Rabbits

Figure 5 shows optical micrographs (Villanueva–Goldner staining) of new bone formed around the Ti–15–4–4 dental implant. New bone was formed at the threaded portion, and the amount of new bone formed increased with the implantation period. 

Figure 6 shows the bone formation on the Ti–15–4–4 dental implant and Ti–15–4–4 smooth surface rods. It was found that the rate of bone formation on the threaded portion was the same as that on the smooth surface. Thus, the Ti–4–4 alloy is expected to be applicable to dental implants [50,51,52,53,54].

### 3.4. Pullout Properties of Blasted Ti–Zr Alloy after Implantation in Rabbits

The effect of surface modification is commonly evaluated by a withdrawal test after the implantation test using rabbits. The site of implantation into the femur of rabbits was determined considering the size of the test specimen and the reactivity between the bone and the implant. The rabbit implantation test was conducted with the implantation location referred to in the literature [55,56,57,58,59,60,61]. Figure 7 shows the changes in maximum pullout load after rabbit implantation as a function of implantation period. The maximum pullout loads of the grit-blasted, shot-blasted, and machined Ti–15–4, Ti–15–4–4, Ti–6–7, and Ti–6–4 alloys increased linearly with implantation period. The pullout load of the grit-blasted Ti–15–4 alloy rods was higher than that of the shot-blasted ones. The increase in maximum pullout load due to the increase in *Ra* was consistent with the trends reported in the literature [62,63,64]. 

Figure 8 shows SEM images of the surfaces of the Ti–15–4 alloy grit-blasted with 24-grit Fuji Random WA Al_2_O_3_ particles and the Alloclassic stem grit-blasted with 24-grit corundum Al_2_O_3_ particles. The blasted Ti–15–4 alloy and Alloclassic stem surfaces appeared similar, as shown in Figure 8a,c. The *Ra* values of the blasted Ti–15–4 and Alloclassic stem surfaces were 3.4 ± 0.1 and 2.9 ± 0.1 μm, respectively. The residual Al_2_O_3_ particles appeared black in the reflected electron image, as indicated by arrows in Figure 8b,d. The area ratio of residual Al_2_O_3_ particles was measured by analyzing six SEM images. The area ratios of residual Al_2_O_3_ particles of the grit-blasted Ti–15–4 and Alloclassic stem surfaces were 9.1 ± 0.4 and 10.4 ± 1.6%, respectively. Residual Al_2_O_3_ particles (surface contamination) of approximately 12 to 23% were observed at the surface of the corundum grit-blasted hip stem [65]. The diameters of the residual Al_2_O_3_ particles ranged from 4 to 100 μm, and 154 particles were counted per mm^2^ [66]. A blasted implant surface had a highly osteoconductive nature [28,55,67]. The area ratio of these residual Al_2_O_3_ particles was close to that reported for the grit-blasted Ti alloy [65]. 

To investigate the effect of pickling treatment on the removal of residual Al_2_O_3_ particles, the grit-blasted Ti–14–4 alloy surface was chemically etched in 2 vol% nitric acid aqueous solution containing 1 vol% hydrogen fluoride for a maximum of up to 4 min. Figure 9 shows changes in the area ratio of residual Al_2_O_3_ particles and *Ra* as functions of immersion time in hydrofluoric acid aqueous solution. The zero on the horizontal axis of Figure 9 represents the state after ultrasonic cleaning of the grit-blasted Ti–15–4 alloy. As the immersion time increased, the area ratio of residual Al_2_O_3_ particles initially increased and then decreased. The increase is considered to be due to the released matrix around the Al_2_O_3_ particles. On the other hand, the change in *Ra* caused by the increase in immersion time was small. It is considered that this alloy could be used for artificial hip joint stems even with ultrasonic cleaning after grit blasting. The sensitization (maximization) test using the accelerated-condition extract of the ultrasonic-cleaned Ti–15–4 alloy after grit blasting showed negative results. When the grit-blasted Ti–14–4 alloy surface was chemically etched in 4 vol% nitric acid aqueous solution containing 2 vol% hydrogen fluoride for 2 min, the area ratio of residual Al_2_O_3_ particles decreased to 0.2 ± 0.1% (*Ra*, 2.3± 0.2 μm). The maximum pullout loads after these surface-treated Ti–15–4 alloys were implanted into eight rabbits for 4, 8, and 12 weeks were 165 ± 45, 153 ± 30, and 197 ± 48 N, respectively.

## 4. Conclusions

To obtain basic data required for the development of orthopedic Ti–Zr alloy implant devices, biological safety evaluation tests of Ti–Zr alloys in accordance with the ISO 10993 series were performed under both normal and accelerated (exaggerated) extraction conditions. Furthermore, to compare the osteocompatibility of implantable alloys and investigate the effects of metal ions, we conducted histopathological analysis with morphometrical parameters for the long-term implantation of pure V, Al, Ni, Zr, Nb, and Ta metals as well as Ni–Ti and high-V-containing Ti–15V–3Al–3Sn alloys in rats. The effect of the dental implant (screw) shape on morphometrical parameters was investigated using rabbits. Moreover, we examined the maximum pullout properties of grit-blasted Ti–Zr alloys after the implantation in rabbits.
The biological safety evaluation tests with the three Ti–Zr alloys in accordance with the ISO 10993 series showed no adverse effects (negative) of either normal or accelerated extraction. A method of evaluating biological safety under accelerated extraction condition is useful for evaluating the biological safety of highly biocompatible materials, biodegradable materials, and so on.No bone was formed around the pure V and Ni implants. The Al, Zr, Nb, and Ni–Ti implants were surrounded by new bone. The rates of new bone formation around pure Zr, Nb, and Ta implants were all approximately 100% in the first four weeks of the implantation period. The rates of bone contact were higher with pure Zr than with Nb and Ta. The new bone thicknesses for pure Zr, Nb, and Ta implants were approximately 40 μm 12 weeks after implantation. The new bone surrounding Ti–Ni and high-V-containing Ti–15–3–3 alloys tended to be thinner than that surrounding Ti–15–4–4 and Ti–6–4 alloys.The rate of bone formation on the threaded portion in the Ti–15–4–4 dental implant was the same as that on the smooth surface. The Ti–4–4 alloy is expected to be applicable to dental implants because it induces excellent bone formation.The maximum pullout loads of the grit-blasted and shot-blasted Ti–15–4 and Ti–15–4–4 alloys increased linearly with the implantation period in rabbits. The pullout load of grit-blasted Ti–15–4 alloy rods was higher than that of shot-blasted ones.The *Ra* of the Ti–15–4 alloy surface grit-blasted with 24-grit Fuji Random WA Al_2_O_3_ particles was the same as that of the grit-blasted Alloclassic stem surface. The *Ra* and area ratio of residual Al_2_O_3_ particles were approximately the same. It was clarified that the grit-blasted Ti–15–4 alloy could be used for artificial hip joint stems. The sensitization (maximization) tests with the accelerated-condition extract of the grit-blasted Ti–15–4 alloy showed negative results.

## Figures and Tables

**Figure 1 materials-14-00731-f001:**
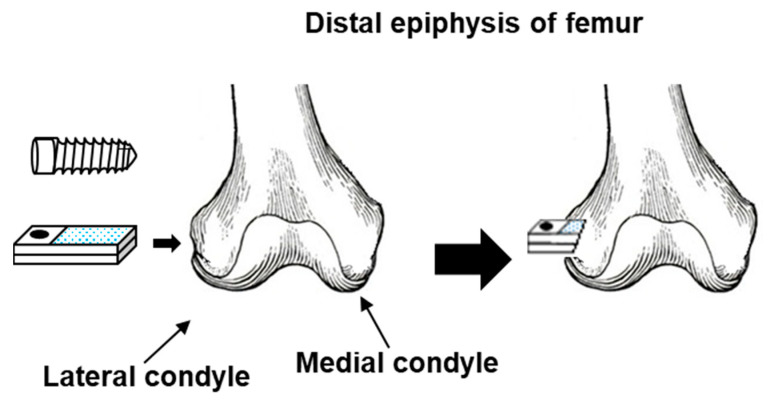
Schematic illustration of method of implantation into femur of rabbits.

**Figure 2 materials-14-00731-f002:**
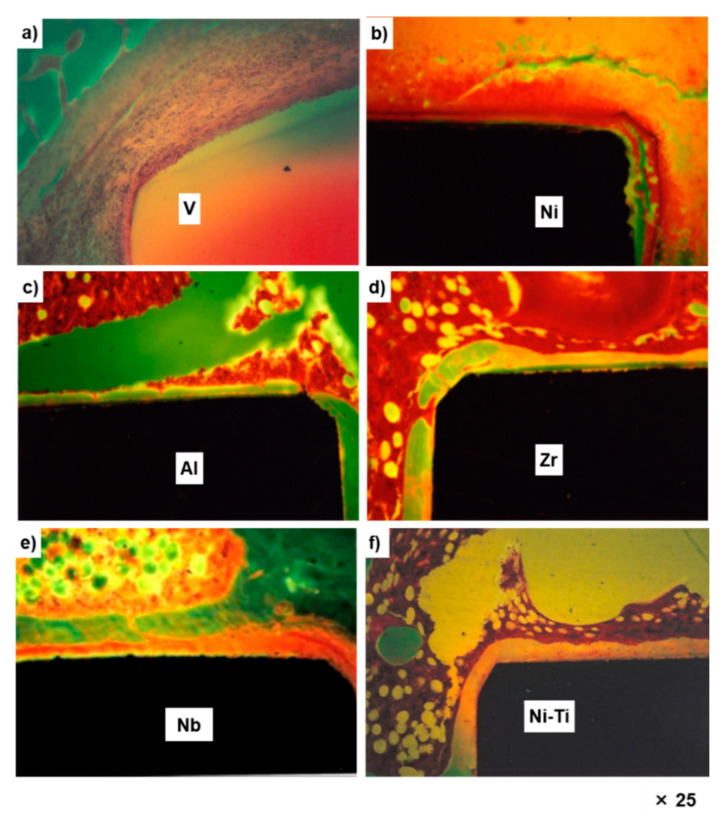
Optical micrographs of new bone formed around pure V (**a**), Ni (**b**), Al (**c**), Zr (**d**), and Nb (**e**) metals and Ni–Ti alloy (**f**) after 52 weeks. Black shadows show implanted metals. For V, the implanted specimen was lost during the preparation of the section.

**Figure 3 materials-14-00731-f003:**
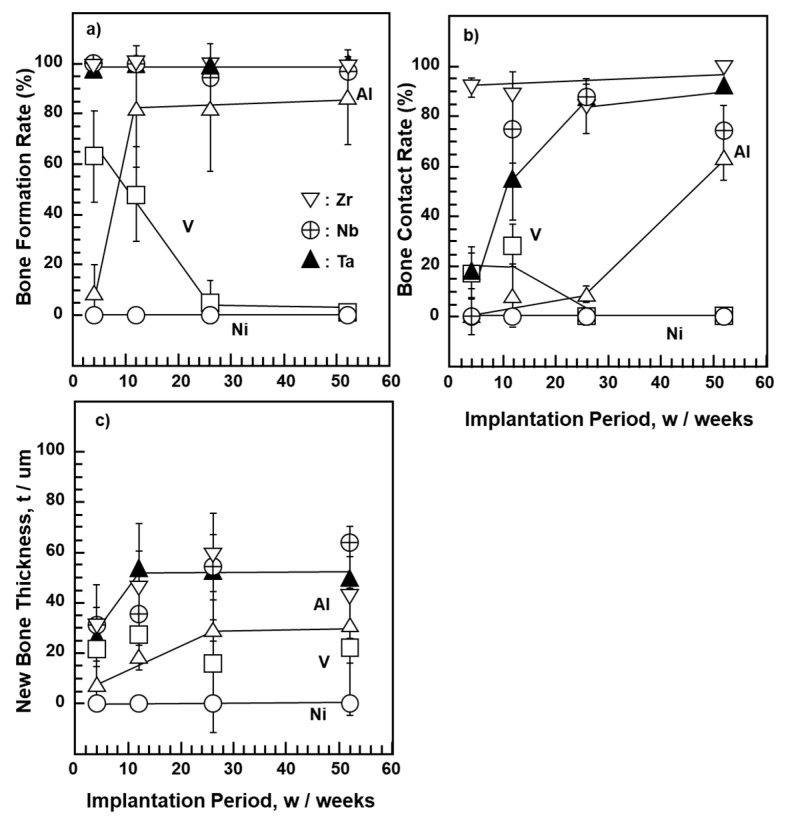
New bone formation rate (**a**), bone contact rate (**b**), and new bone thickness (**c**) for pure V, Ni, Al, Zr, and Nb.

**Figure 4 materials-14-00731-f004:**
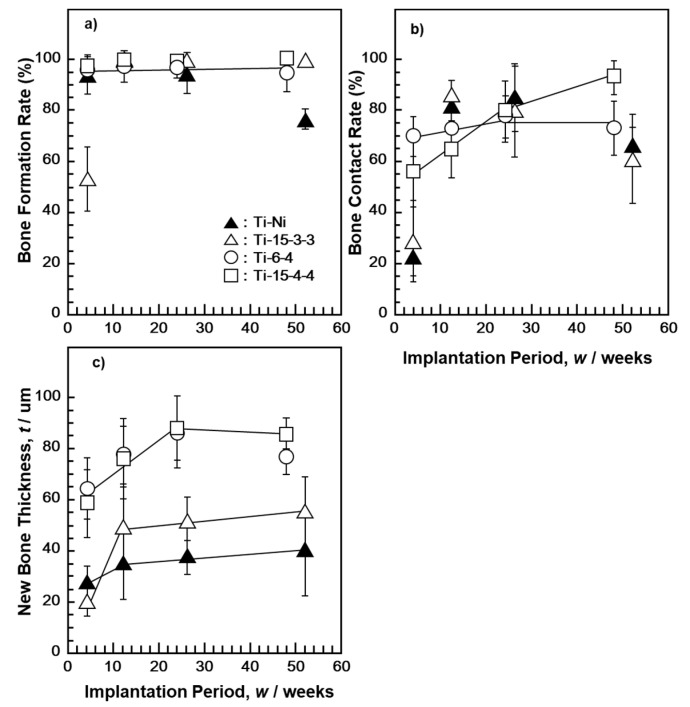
New bone formation rate (**a**), bone contact rate (**b**), and new bone thickness (**c**) for Ti–Ni, Ti–15–3–3, Ti–6–4 [16], and Ti–15–4–4 [16].

**Figure 5 materials-14-00731-f005:**
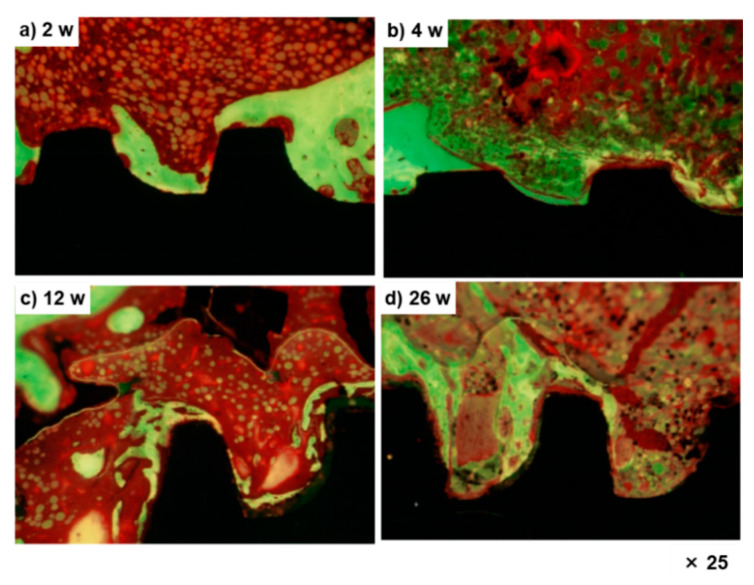
Optical micrographs of new bone formed around Ti–15–4–4 dental implant after 2 (**a**), 4 (**b**), 12 (**c**), and 26 (**d**) weeks.

**Figure 6 materials-14-00731-f006:**
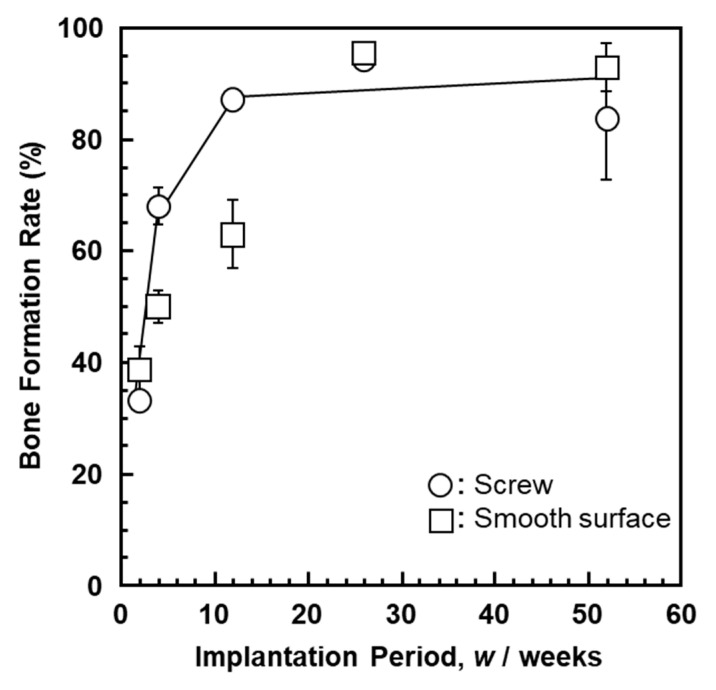
New bone formation rates for Ti–15–4–4 dental implant and Ti–15–4–4 smooth surface rods.

**Figure 7 materials-14-00731-f007:**
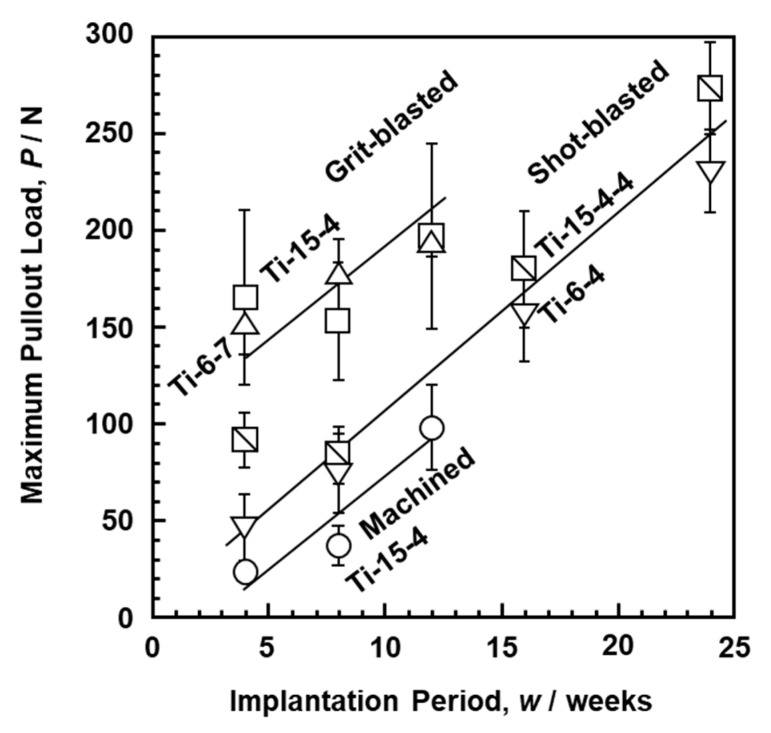
Maximum pullout loads after implantation in rabbits.

**Figure 8 materials-14-00731-f008:**
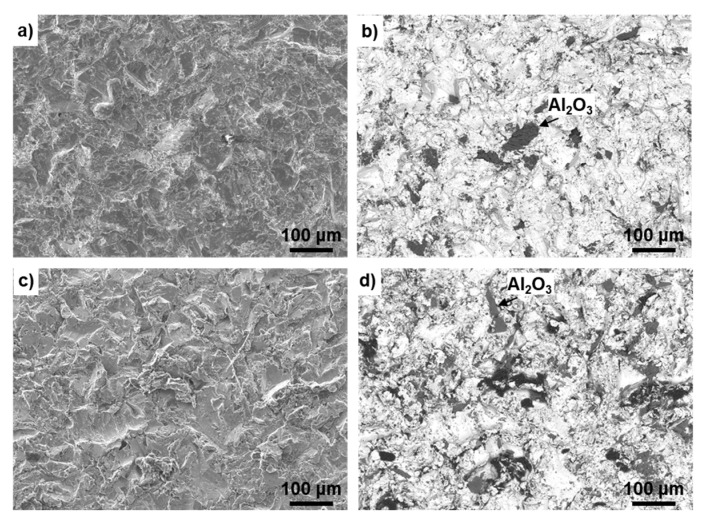
SEM images of grit-blasted surfaces. (**a**,**b**) Ti–15–4 alloy surface blasted with 24-grit Fuji Random WA Al_2_O_3_ particles, (**c**,**d**) Alloclassic stem surfaces blasted with 24-grit corundum Al_2_O_3_ particles, (**a**,**c**) secondary electron images, and (**b**,**d**) reflected electron images.

**Figure 9 materials-14-00731-f009:**
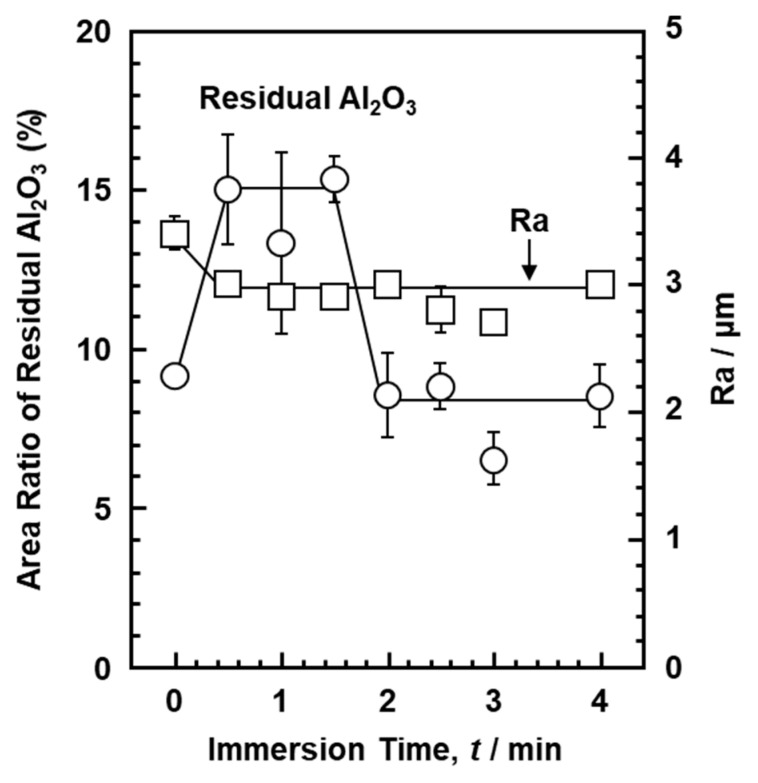
Changes in area ratio of residual Al_2_O_3_ and surface roughness (*Ra*) as functions of immersion time in hydrofluoric acid aqueous solution.

**Table 1 materials-14-00731-t001:** Chemical compositions (mass%) of the three Ti–Zr alloys used.

**Alloy**	**Zr**	**Nb**	**Ta**	**Pd**	**Fe**	**O**	**N**	**H**	**C**	**Ti**
Ti–15–4	16.10	3.90	0.17	<0.01	0.026	0.254	0.080	0.0010	0.010	Bal.
Ti–15–4–1	17.24	3.97	1.67	0.02	0.036	0.29	0.096	0.0027	0.011	Bal.
Ti–15–4–4	16.55	4.0	3.9	<0.01	0.04	0.28	0.09	0.0012	0.007	Bal.
	**Al**	**V**	**Cr**	**Sn**	**Fe**	**O**	**N**	**H**	**C**	**Ti**
Ti–6–4	6.4	4.4	−	−	0.10	0.07	0.02	0.0027	0.025	Bal.
Ti–15–3–3	3.14	15.285	3.13	3.06	0.22	0.109	0.0085	0.013	−	Bal.

**Table 2 materials-14-00731-t002:** Test conditions of biological safety evaluation used in this study.

Evaluation Item	Test System	Test Specimen	Extraction
1. Cytotoxicity Colony formation <ISO 10993-5 [19]>	Cultured cells (V79)	4 Disks (3.5 mm diameter, 1 mm thickness)	(1) Direct contact
		10 Plates (2 cm × 2 cm × 1 mm)	(2) Extraction in culture medium (37 °C for 24 h)Extraction rate: 6 cm^2^/mL
2. Sensitization GPMT <ISO 10993-10 [20]>	Guinea pigs	9 Plates	Extraction rate: 3 cm^2^/mL (1) Accelerated extraction (0.9%NaCl + HCl, pH = 2) Neutralized with NaOH after extraction at 37 °C for 7 d (2) Extraction in 0.9%NaCl (121 °C for 1 h)
3. Irritation Intracutaneous reactivity <ISO 10993-10 [20]>	Rabbits	6 Plates	
4. Systemic toxicity <ISO 10993-11 [21]> (a) Acute systemic toxicity	Mice	8 Plates	
(b) Subacute systemic toxicity	Rats (21-day intravenous administration)	51 Plates (2 cm×2 cm×1 mm)	Extraction in 0.9%NaCl (121 °C for 1 h) Extraction rate: 3 cm^2^/mL
5. Genotoxicity <ISO 10993-3 [22] ISO 10993-33 [23]> (a) Reverse mutation	4 strains of *Salmonella typhimurium* and *Escherichia coli*	16 Plates (2 cm×2 cm×1 mm)	Extraction rate: 6 cm^2^/mL Accelerated extraction
(b) Chromosomal abnormalities	Cultured cells (CHL/IU)	120 Plates (2 cm × 2 cm × 1 mm)	Extraction in medium(37 °C for 48 h)
6. Implantation <ISO 10993-6 [24]>	Rats	24 Circular rods (1.2 mm diameter, 2.5 mm length)	Femur diaphysis 3- and 6-month implantations

GPMT: Guinea pig maximization test.

**Table 3 materials-14-00731-t003:** Results of biological safety evaluation in accordance with the ISO 10993 series.

Evaluation Item	Ti–15–4	Ti–15–4–1	Ti–15–4–4
1. Cytotoxicity Colony formation	Medium extract: noncytotoxic	Medium extract: noncytotoxic	Direct contact: noncytotoxic
2. Sensitization	Accelerated extract: negative	Accelerated extract: negative	0.9%NaCl extract: negative
3. Irritation Intracutaneous reactivity	Accelerated extract: negative 0.9%NaCl extract: negative	0.9%NaCl extract: negative	
4. Systemic toxicity (a) Acute systemic toxicity (b) Subacute systemic toxicity	Accelerated extract: nontoxic 0.9%NaCl extract: nontoxic 0.9%NaCl extract: nontoxic	0.9%NaCl extract: nontoxic 0.9%NaCl extract: nontoxic	
5. Genotoxicity (a) Reverse mutation (b) Chromosomal abnormality	Accelerated extract: negative Medium extract: negative	0.9%NaCl extract: negative Medium extract: negative	0.9%NaCl extract: negative Medium extract: negative
6. Implantation	No inflammation in local implanted region, bone conductivity equivalent to control specimen	No inflammation in local implanted region, bone conductivity equivalent to control specimen	No inflammation in local implanted region, bone conductivity equivalent to control specimen

## Data Availability

Data sharing is not applicable to this article.

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
