# Peer review of "Biological Safety Evaluation and Surface Modification of Biocompatible Ti–15Zr–4Nb Alloy"

_materials, 2021, doi:10.3390/ma14040731_

Round 1
Reviewer 1 Report
The manuscript "Biological Safety Evaluation and Surface
Modification of Biocompatible Ti-15Zr-4Nb Alloy" is well written and the research methodology and the results are clearly presented.
Such studies are of real interest
Author Response
Thank you for the peer review of our manuscript.
Reviewer 2 Report
Ther paper reports different safety tests of Ti-Zr alloys as requested by ISO 12 10993 series for the biological evaluation of medical devices.
The study is well structured and data are presented in a clear way.
I would suggest to summarize the introduction, focusing more on the applications in orthopedic implants and on osseointegration.
As for the methods, more information should be provided on animal care and monitoring. Please check ARRIVE guidelines (https://arriveguidelines.org/) on this issue.
Minor corrections:
- Various lines: please substitute "artificial tooth roots" or "artificial tooth root scew" with "dental implants".
- Line 508. A reference should be added after "induces high-rate bone formation" (see for istance:PMID: 26800182).
- Figure 4 reports non-original data. The authors should consider whether it is necessary to include these data as a figure or simply discuss it in the maintext. However, reference should be reported also in the figure's caption.
- Figure 5 caption: staining must be reported.
Author Response
Thank you for the peer review of the manuscript, which has been revised in accordance with your comments.
Introduction was shortened to focus on orthopedic implants and bone formation. The evaluation parameters for bone formation have been moved to the end of 2.3 (6) Implantation tests. The explanation for ISO 10993 has been shortened but left as important because it is important to consider the accelerated test method.
With reference to the ARRIVE guidance, animal care and monitoring were added to 2.2. Animals, cells, and bacteria. Also, the description of the preliminary test has been deleted to shorten the description of the test method.
We have modified to “artificial tooth roots” and “artificial tooth root screw” “dental implants”.
References have been added as citations after "induces high-rate bone formation"; however, we have revised the text because we cannot state this from this result alone.
Regarding Fig. 4, the data of Ti-6-4 and Ti-15-4-4 were taken from our previous studies, but the data of the two other materials (Ti-Ni and Ti-15-3-3) are original ones. References are cited in captions forTi-6-4 and Ti-15-4-4.
The staining method was added to the explanation in Fig. 5.
Reviewer 3 Report
The article submitted for review, entitled "Biological Safety Evaluation and Surface 3 Modification of Biocompatible Ti-15Zr-4Nb Alloy" does not meet scientific articles' basic requirements. The volume of introduction and experiment section accounts for more than half of the article while containing a lot of irrelevant information. The presented research results have not been subject to discussion by reference to the existing literature reports. In addition, I would like to point out glaring irregularities in the literature cited, i.e., the main author of this article cites 11 of her own works, 8 ISO standards out of a total of 36 items. Moreover, the manuscript contains numerous fragments that are plagiarism from other articles.
Author Response
Thank you for the peer review of the manuscript, which has been revised in accordance with your comments.
Since there have been no studies in which all of the items in the ISO 10993 series (i.e., cytotoxicity, sensitization, irritation/intracutaneous reactivity, acute systemic toxicity, subacute toxicity, genotoxicity, and implantation) were evaluated, we conducted biological safety evaluations using Ti-Zr alloys in accordance with the ISO 10993 series. In addition to the conventional method, we have developed an accelerated extraction method to improve sensitivity. Regarding the descriptions of test methods that are lengthy, we have made them as concise as possible but still understandable. The related literature has been reviewed and revised. The modifications are as follows:
(1) We have reduced the number of fragments from other articles and made the introduction as short as possible.
(2) We have increased the relevant citations and added discussion to the test results.
(3) We have corrected the descriptions so that they are consistent.
The ISO 10993 series have a lot of test items, which makes the test method longer, however, we believe that readers will be understand it.
Round 2
Reviewer 3 Report
I regret to say that the authors neglected to very serious allegations presented in my previous review or dealt with them in a very careless manner.
1. It was necessary to reduce the paper 8-16 (self-citation) to a maximum of 3 because most of them are cited in the manuscript one time in a very overall sentence: "Ti-15Zr-48 4Nb-(0 to 4) Ta alloy, which is a Ti-Zr (α-β-type) alloy, has been developed in Japan as a highly biocompatible alloy for long-term biomedical application [8–16]".
2. According to the author's declaration: "(2) We have increased the relevant citations and added a discussion to the test results." I have to challenge this statement. The discussion was extended just with 3 fragments, i.e. 442-450, 500-503, 516-520, which simultaneously with the expansion of the experimental part in no way constitutes an extension of the discussion. Additionally, added text "There were studies in which some of the biological safety items for Ti-6Al-4V and Ti-Ni alloys 442 were evaluated [33–38]." presents excessive and incorrect citations - these works are not discussed in any way and add nothing to the discussion. This error is repeated many times in the text.
3. Introduction and experimental section were not changed, and my suggestions were completely omitted.
Author Response
Thank you for the peer review of the manuscript again. We were not fully aware of the length restrictions on the description of the test method. We also did not understand the limits on the percentage of self-citations. Owing to our inadequate understanding, we caused a great deal of inconvenience this time. We have fully taken these into consideration and carefully revised the entire manuscript it again. The corrections are as follows.
- References 8-16 (self-citation)
The treatise has been deleted. Also, the related sentences have been shortened. The entire introduction has been revised and shortened. The cited references have also been reviewed and amended as appropriate.
About discussion
We reconfirmed the literature related to the results of this test so as not to violate the points pointed out. We have corrected the discussion in “3. Experimental results”. The poor use of English may have led to this misunderstanding. We hope that you understand that we are not simply quoting unrelated literature.
- Introduction and Experimental method
(a) The “Introduction” has been shortened.
(b) The section “2.3. Biological safety evaluation tests” has been shortened. The explanation for each test item has been deleted. Instead, Table 1 cites and briefly summarizes the ISO test methods.
(c) Also, the section “2.2. Animals, cells, and bacteria” has been added as pointed out by Reviewer 2
